# Natasha 2: Faster Non-Convex Optimization Than SGD

**Zeyuan Allen-Zhu**[*]
Microsoft Research AI
zeyuan@csail.mit.edu

## Abstract

We design a stochastic algorithm to find $\varepsilon$-approximate local minima of any smooth nonconvex function in rate $O(\varepsilon^{-3.25})$, with only oracle access to stochastic gradients. The best result before this work was $O(\varepsilon^{-4})$ by stochastic gradient descent (SGD).[2]

## 1 Introduction

In diverse world of deep learning research has given rise to numerous architectures for neural networks (convolutional ones, long short term memory ones, etc). However, to this date, the underlying training algorithms for neural networks are still stochastic gradient descent (SGD) and its heuristic variants. In this paper, we address the problem of designing a new algorithm that has provably faster running time than the best known result for SGD.

Mathematically, we study the problem of online stochastic nonconvex optimization:

$$\min_{x \in \mathbb{R}^d} \left\{ f(x) := \mathbb{E}_i[f_i(x)] = \tfrac{1}{n} \textstyle\sum_{i=1}^n f_i(x) \right\} \tag{1.1}$$

where both $f(\cdot)$ and each $f_i(\cdot)$ can be nonconvex. We want to study

*online algorithms* to find appx. *local minimum* of $f(x)$.

Here, we say an algorithm is online if its complexity is independent of $n$. This tackles the big-data scenarios when $n$ is extremely large or even infinite.[3]

Nonconvex optimization arises prominently in large-scale machine learning. Most notably, training deep neural networks corresponds to minimizing $f(x)$ of this average structure: each training sample $i$ corresponds to one loss function $f_i(\cdot)$ in the summation. This average structure allows one to perform stochastic gradient descent (SGD) which uses a random $\nabla f_i(x)$ —corresponding to computing backpropagation once— to approximate $\nabla f(x)$ and performs descent updates.

The standard goal of nonconvex optimization with provable guarantee is to find *approximate local minima*. This is not only because finding the *global* one is NP-hard, but also because there exist rich literature on *heuristics* for turning a local-minima finding algorithm into a global one. This includes random seeding, graduated optimization [25] and others. Therefore, faster algorithms for finding approximate local minima translate into faster *heuristic* algorithms for finding global minimum.

On a separate note, experiments [16, 17, 24] suggest that fast convergence to approximate local minima may be sufficient for training neural nets, while convergence to stationary points (i.e., points that may be saddle points) is *not*. In other words, we need to *avoid saddle points*.

---

[*]The full version of this paper can be found on https://arxiv.org/abs/1708.08694.

[2]When this manuscript first appeared online, the best rate was $T = O(\varepsilon^{-4})$ by SGD. Several followups appeared after this paper. This includes stochastic cubic regularization [44] which gives $T = O(\varepsilon^{-3.5})$, and Neon+SCSG [10, 46] which gives $T = O(\varepsilon^{-3.333})$. These rates are worse than $T = O(\varepsilon^{-3.25})$.

[3]All of our results in this paper apply to the case when $n$ is infinite, meaning $f(x) = \mathbb{E}_i[f_i(x)]$, because we focus on *online* methods. However, we still introduce $n$ to simplify notations.

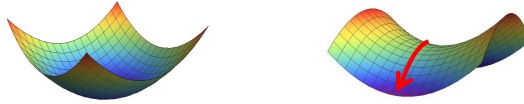

Figure 1: Local minimum (left), saddle point (right) and its negative-curvature direction.

## 1.1 Classical Approach: Escaping Saddle Points Using Random Perturbation

One natural way to avoid saddle points is to use randomness to escape from it, whenever we meet one. For instance, Ge et al. [22] showed, by injecting random perturbation, SGD will not be stuck in saddle points: whenever SGD moves into a saddle point, randomness shall help it escape. This partially explains why SGD performs well in deep learning.[4] Jin et al. [27] showed, equipped with random perturbation, full gradient descent (GD) also escapes from saddle points. Being easy to implement, however, we raise two main efficiency issues regarding this classical approach:

- Issue 1. If we want to escape from saddle points, is random perturbation the only way? Moving in a random direction is "blind" to the Hessian information of the function, and thus can we escape from saddle points faster?

- Issue 2. If we want to avoid saddle points, is it really necessary to first move close to saddle points and then *escape* from them? Can we design an algorithm that can somehow avoid saddle points without ever moving close to them?

## 1.2 Our Resolutions

**Resolution to Issue 1: Efficient Use of Hessian.**   Mathematically, instead of using a random perturbation, the negative eigenvector of $\nabla^2 f(x)$ (a.k.a. the negative-curvature direction of $f(\cdot)$ at $x$) gives us a *better* direction to escape from saddle points. See Figure 1.

To make it concrete, suppose we apply power method on $\nabla^2 f(x)$ to find its most negative eigenvector. If we run power method for 0 iteration, then it gives us a totally random direction; if we run it for more iterations, then it converges to the most negative eigenvector of $\nabla^2 f(x)$. Unfortunately, applying power method is unrealistic because $f(x) = \frac{1}{n} \sum_i f_i(x)$ can possibly have infinite pieces.

We propose to use Oja's algorithm [37] to approximate power method. Oja's algorithm can be viewed as an online variant of power method, and requires only (stochastic) matrix-vector product computations. In our setting, this is the same as (stochastic) Hessian-vector products —namely, computing $\nabla^2 f_i(x) \cdot w$ for arbitrary vectors $w \in \mathbb{R}^d$ and random indices $i \in [n]$. It is a known fact that computing Hessian-vector products is as cheap as computing stochastic gradients, and thus we can use Oja's algorithm to escape from saddle points.

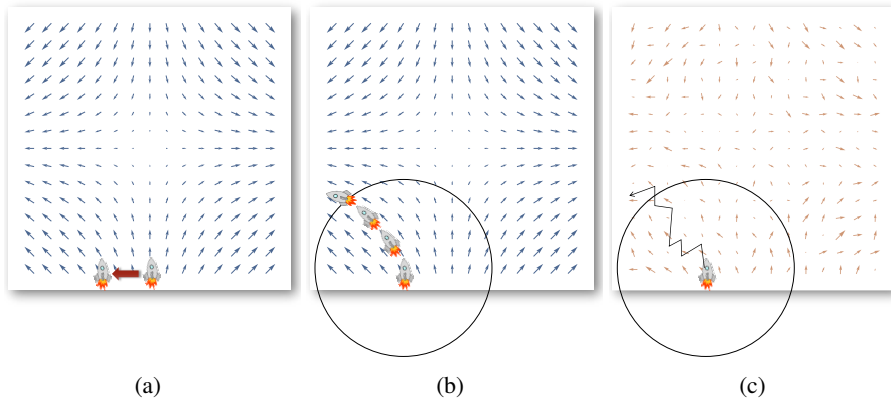

(a)                                    (b)                                    (c)

Figure 2: Illustration of `Natasha2`$^{\text{full}}$ — how to swing by a saddle point.
     (a) move in a negative curvature direction if there is any (by applying Oja's algorithm)
     (b) swing by a saddle point without entering its neighborhood (wishful thinking)
     (c) swing by a saddle point using only stochastic gradients (by applying `Natasha1.5`$^{\text{full}}$)

**Resolution to Issue 2: Swing by Saddle Points.** If the function is sufficiently smooth,[5] then any point close to a saddle point must have a negative curvature. Therefore, as long as we are close to saddle points, we can already use Oja's algorithm to find such negative curvature, and move in its direction to decrease the objective, see Figure 2(a).

Therefore, we are left only with the case that point is not close to any saddle point. Using smoothness of $f(\cdot)$, this gives a "safe zone" near the current point, in which there is no strict saddle point, see Figure 2(b). Intuitively, we wish to use the property of safe zone to design an algorithm that decreases the objective faster than SGD. Formally, $f(\cdot)$ inside this safe zone must be of "bounded nonconvexity," meaning that its eigenvalues of the Hessian are always greater than some negative threshold $-\sigma$ (where $\sigma$ depends on how long we run Oja's algorithm). Intuitively, the greater $\sigma$ is, then the more non-convex $f(x)$ is. We wish to design an (online) stochastic first-order method whose running time scales with $\sigma$.

Unfortunately, classical stochastic methods such as SGD or SCSG [30] cannot make use of this nonconvexity parameter $\sigma$. The only known ones that can make use of $\sigma$ are offline algorithms. In this paper, we design a new stochastic first-order method `Natasha1.5`

**Theorem 1** (informal). `Natasha1.5` *finds $x$ with $\|\nabla f(x)\| \leq \varepsilon$ in rate $T = O\big(\frac{1}{\varepsilon^3} + \frac{\sigma^{1/3}}{\varepsilon^{10/3}}\big)$* .

Finally, we put `Natasha1.5` together with Oja's to construct our final algorithm `Natasha2`:

**Theorem 2** (informal). `Natasha2` *finds $x$ with $\|\nabla f(x)\| \leq \varepsilon$ and $\nabla^2 f(x) \succeq -\delta\mathbf{I}$ in rate $T = \widetilde{O}\big(\frac{1}{\delta^5} + \frac{1}{\delta\varepsilon^3} + \frac{1}{\varepsilon^{3.25}}\big)$. In particular, when $\delta \geq \varepsilon^{1/4}$, this gives $T = \widetilde{O}\big(\frac{1}{\varepsilon^{3.25}}\big)$.*

In contrast, the convergence rate of SGD was $T = \widetilde{O}(\mathsf{poly}(d) \cdot \varepsilon^{-4})$ [22].

## 1.3 Follow-Up Results

Since the original appearance of this work, there has been a lot of progress in stochastic nonconvex optimization. Most notably,

- If one *swings by* saddle points using Oja's algorithm and SGD variants (instead of `Natasha1.5`), the convergence rate is $T = \widetilde{O}(\varepsilon^{-3.5})$ [5].
- If one applies SGD and only *escapes from* saddle points using Oja's algorithm, the convergence rate is $T = \widetilde{O}(\varepsilon^{-4})$ [10, 46].
- If one applies SCSG and only *escapes from* saddle points using Oja's algorithm, the convergence rate is $T = \widetilde{O}(\varepsilon^{-3.333})$ [10, 46].
- If one applies a stochastic version of cubic regularized Newton's method, the convergence rate is $T = \widetilde{O}(\varepsilon^{-3.5})$ [44].
- If $f(x)$ is of $\sigma$-bounded nonconvexity, the SGD4 method [5] gives rate $T = \widetilde{O}(\varepsilon^{-2} + \sigma\varepsilon^{-4})$.

We include these results in Table 1 for a close comparison.

## 2 Preliminaries

Throughout this paper, we denote by $\|\cdot\|$ the Euclidean norm. We use $i \in_R [n]$ to denote that $i$ is generated from $[n] = \{1, 2, \ldots, n\}$ uniformly at random. We denote by $\nabla f(x)$ the gradient of function $f$ if it is differentiable, and $\partial f(x)$ any subgradient if $f$ is only Lipschitz continuous. We denote by $\mathbb{I}[event]$ the indicator function of probabilistic events.

We denote by $\|\mathbf{A}\|_2$ the spectral norm of matrix $\mathbf{A}$. For symmetric matrices $\mathbf{A}$ and $\mathbf{B}$, we write $\mathbf{A} \succeq \mathbf{B}$ to indicate that $\mathbf{A} - \mathbf{B}$ is positive semidefinite (PSD). Therefore, $\mathbf{A} \succeq -\sigma\mathbf{I}$ if and only if all eigenvalues of $\mathbf{A}$ are no less than $-\sigma$. We denote by $\lambda_{\min}(\mathbf{A})$ and $\lambda_{\max}(\mathbf{A})$ the minimum and maximum eigenvalue of a symmetric matrix $\mathbf{A}$.

**Definition 2.1.** *For a function $f : \mathbb{R}^d \to \mathbb{R}$,*

- *$f$ is $\sigma$-strongly convex if $\forall x, y \in \mathbb{R}^d$, it satisfies $f(y) \geq f(x) + \langle \partial f(x), y - x \rangle + \frac{\sigma}{2}\|x - y\|^2$.*

| | algorithm | gradient complexity $T$ | variance bound | Lipschitz smooth | 2nd-order smooth |
|---|---|---|---|---|---|
| convex only | SGD1 [5, 23] | $O(\varepsilon^{-2.667})$ | needed | needed | no |
| | SGD2 [5] | $O(\varepsilon^{-2.5})$ ♮ | needed | needed | no |
| | SGD3 [5] | $\widetilde{O}(\varepsilon^{-2})$ ♮ | needed | needed | no |
| approximate stationary points | SGD (folklore) | $O(\varepsilon^{-4})$ (see Appendix B) | needed | needed | no |
| | SCSG [30] | $O(\varepsilon^{-3.333})$ | needed | needed | no |
| | Natasha1.5 | $O(\varepsilon^{-3} + \sigma^{1/3}\varepsilon^{-3.333})$ (see Theorem 1) | needed | needed | no |
| | SGD4 [5] | $\widetilde{O}(\varepsilon^{-2} + \sigma\varepsilon^{-4})$ ♮ | needed | needed | no |
| approximate local minima | perturbed SGD [22] | $\widetilde{O}(\varepsilon^{-4} \cdot \mathsf{poly}(d))$ | needed | needed | needed |
| | Natasha2 | $\widetilde{O}(\varepsilon^{-3.25})$ (see Theorem 2) | needed | needed | needed |
| | NEON + SGD [10, 46] | $\widetilde{O}(\varepsilon^{-4})$ ♮ | needed | needed | needed |
| | cubic Newton [44] | $\widetilde{O}(\varepsilon^{-3.5})$ ♮ | needed | needed | needed |
| | SGD5 [5] | $\widetilde{O}(\varepsilon^{-3.5})$ ♮ | needed | needed | needed |
| | NEON + SCSG [10, 46] | $\widetilde{O}(\varepsilon^{-3.333})$ ♮ | needed | needed | needed |

Table 1: Comparison of **_online_** methods for finding $\|\nabla f(x)\| \leq \varepsilon$. Following tradition, in these complexity bounds, we assume variance and smoothness parameters as constants, and only show the dependency on $n, d, \varepsilon$ and the bounded nonconvexity parameter $\sigma \in (0, 1)$. We use ♮ to indicate results that appeared after this paper.

---

**Remark 1.** Variance bounds must be needed for online methods.

**Remark 2.** Lipschitz smoothness must be needed for achieving even approximate stationary points.

**Remark 3.** Second-order smoothness must be needed for achieving approximate local minima.

- $f$ is of $\sigma$-bounded nonconvexity (or $\sigma$-**_nonconvex_** for short) if $\forall x, y \in \mathbb{R}^d$, it satisfies $f(y) \geq f(x) + \langle \partial f(x), y - x \rangle - \frac{\sigma}{2}\|x - y\|^2$. [6]
- $f$ is $L$-Lipschitz smooth (or $L$-**_smooth_** for short) if $\forall x, y \in \mathbb{R}^d$, $\|\nabla f(x) - \nabla f(y)\| \leq L\|x - y\|$.
- $f$ is second-order $L_2$-Lipschitz smooth (or $L_2$-**_second-order smooth_** for short) if $\forall x, y \in \mathbb{R}^d$, it satisfies $\|\nabla^2 f(x) - \nabla^2 f(y)\|_2 \leq L_2\|x - y\|$.

These definitions have other equivalent forms, see textbook [33].

**Definition 2.2.** _For composite function $F(x) = \psi(x) + f(x)$ where $\psi(x)$ is proper convex, given a parameter $\eta > 0$, the_ **_gradient mapping_** _of $F(\cdot)$ at point $x$ is_

$$\mathcal{G}_{F,\eta}(x) := \frac{1}{\eta}(x - x') \quad where \quad x' = \arg\min_y \left\{ \psi(y) + \langle \nabla f(x), y \rangle + \frac{1}{2\eta}\|y - x\|^2 \right\}$$

_In particular, if $\psi(\cdot) \equiv 0$, then $\mathcal{G}_{F,\eta}(x) \equiv \nabla f(x)$._

## 3 Natasha 1.5: Finding Approximate Stationary Points

We first make a detour to study how to find approximate stationary points using only first-order information. A point $x \in \mathbb{R}^d$ is an $\varepsilon$-approximate stationary point[7] of $f(x)$ if it satisfies $\|\nabla f(x)\| \leq \varepsilon$. Let _gradient complexity $T$_ be the number of computations of $\nabla f_i(x)$.

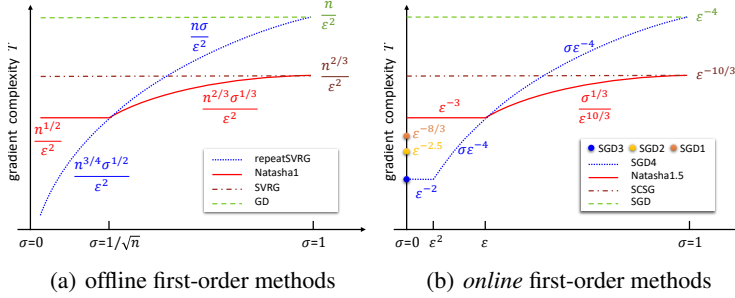

Figure 3: Comparison of first-order methods for finding $\varepsilon$-approximate stationary points of a $\sigma$-nonconvex function. For simplicity, in the plots we let $L = 1$ and $\mathcal{V} = 1$. The results SGD2/3/4 appeared after this work.

Before 2015, nonconvex first-order methods give rise to two convergence rates. SGD converges in $T = O(\varepsilon^{-4})$ and GD converges $T = O(n\varepsilon^{-2})$. The proofs of both are simple (see Appendix B for completeness). In particular, the convergence of SGD relies on two *minimal* assumptions

$$f(x) \text{ has bounded variance } \mathcal{V}, \text{ meaning } \mathbb{E}_i[\|\nabla f_i(x) - \nabla f(x)\|^2] \leq \mathcal{V}, \text{ and} \qquad \text{(A1)}$$

$$f(x) \text{ is } L\text{-Lipschitz smooth, meaning } \|\nabla f(x) - \nabla f(y)\| \leq L \cdot \|x - y\|. \qquad \text{(A2')}$$

*Remark* 3.1. Both assumptions are necessary to design online algorithms for finding stationary points.[8] For *offline* algorithms —like GD— the first assumption is not needed.

Since 2016, the convergence rates have been improved to $T = O(n + n^{2/3}\varepsilon^{-2})$ for offline methods [6, 38], and to $T = O(\varepsilon^{-10/3})$ for online algorithms [30]. Both results are based on the SVRG (stochastic variance reduced gradient) method, and assume additionally (note (A2) implies (A2'))

$$\text{each } f_i(x) \text{ is } L\text{-Lipschitz smooth.} \qquad \text{(A2)}$$

Lei et al. [30] gave their algorithm a new name, SCSG (stochastically controlled stochastic gradient).

**Bounded Non-Convexity.** In recent works [3, 13], it has been proposed to study a more refined convergence rate, by assuming that $f(x)$ is of *$\sigma$-bounded nonconvexity* (or $\sigma$-nonconvex), meaning

$$\text{all the eigenvalues of } \nabla^2 f(x) \text{ lie in } [-\sigma, L] \qquad \text{(A3)}$$

for some $\sigma \in (0, L]$. This parameter $\sigma$ is analogous to the strong-convexity parameter $\mu$ in convex optimization, where all the eigenvalues of $\nabla^2 f(x)$ lie in $[\mu, L]$ for some $\mu > 0$.

In our illustrative process to "swing by a saddle point," the function inside safe zone —see Figure 2(b)— is also of bounded nonconvexity. Since larger $\sigma$ means the function is "more non-convex" and thus harder to optimize, can we design algorithms with gradient complexity $T$ as an *increasing function of $\sigma$* ?

*Remark* 3.2. Most methods (SGD, SCSG, SVRG and GD) do not run faster in theory if $\sigma < L$.

In the *offline* setting, two methods are known to make use of parameter $\sigma$. One is repeatSVRG, implicitly in [13] and formally in [3]. The other is Natasha1 [3]. repeatSVRG performs better when $\sigma \leq L/\sqrt{n}$ and Natasha1 performs better when $\sigma \geq L/\sqrt{n}$. See Figure 3(a) and Table 2.

Before this work, no online method is known to take advantage of $\sigma$.

### 3.1 Our Theorem

We show that, under (A1), (A2) and (A3), one can non-trivially extend Natasha1 to an *online* version, taking advantage of $\sigma$, and achieving better complexity than SCSG.

Let $\Delta_f$ be any upper bound on $f(x_0) - f(x^*)$ where $x_0$ is the starting point. In this section, to present the simplest results, we use the big-$O$ notion to hide dependency in $\Delta_f$ and $\mathcal{V}$. In Section 6,

**Algorithm 1** `Natasha1.5`$(F, x^{\varnothing}, B, T', \alpha)$

---

**Input:** $f(\cdot) = \frac{1}{n}\sum_{i=1}^{n} f_i(x)$, starting vector $x^{\varnothing}$, epoch length $B \in [n]$, epoch count $T' \geq 1$, learning rate $\alpha > 0$.

1: $\widehat{x} \leftarrow x^{\varnothing}$; $\; p \leftarrow \Theta((\sigma/\varepsilon L)^{2/3})$; $\; m \leftarrow B/p$; $\; X \leftarrow []$;
2: **for** $k \leftarrow 1$ **to** $T'$ **do**                            $\diamond$ *$T'$ epochs each of length $B$*
3:      $\widetilde{x} \leftarrow \widehat{x}$; $\mu \leftarrow \frac{1}{B}\sum_{i \in S} \nabla f_i(\widetilde{x})$ where $S$ is a uniform random subset of $[n]$ with $|S| = B$;
4:      **for** $s \leftarrow 0$ **to** $p - 1$ **do**                     $\diamond$ *$p$ sub-epochs each of length $m$*
5:          $x_0 \leftarrow \widehat{x}$; $\; X \leftarrow [X, \widehat{x}]$;
6:          **for** $t \leftarrow 0$ **to** $m - 1$ **do**
7:              $\widetilde{\nabla} \leftarrow \nabla f_i(x_t) - \nabla f_i(\widetilde{x}) + \mu + 2\sigma(x_t - \widehat{x})$ where $i \in_R [n]$
8:              $x_{t+1} = x_t - \alpha\widetilde{\nabla}$;
9:          **end for**
10:          $\widehat{x} \leftarrow$ a random choice from $\{x_0, x_1, \ldots, x_{m-1}\}$;      $\diamond$ *in practice, choose the average*
11:      **end for**
12: **end for**
13: $\widehat{y} \leftarrow$ a random vector in $X$.                         $\diamond$ *in practice, simply return $\widehat{y}$*
14: $g(x) := f(x) + \sigma\|x - \widehat{y}\|^2$ and use convex SGD to minimize $g(x)$ for $T_{\mathsf{sgd}} = T'B$ iterations.
15: **return** $x^{\mathsf{out}} \leftarrow$ the output of SGD.

---

we shall add back such dependency and as well as support the existence of a proximal term. (That is, to minimize $\psi(x) + f(x)$ where $\psi(x)$ is a proper convex simple function.)

Under such simplified notations, our main theorem can be stated as follows.

**Theorem 1** (simple). *Under (A1), (A2) and (A3), using the big-O notion to hide dependency in $\Delta_f$ and $\mathcal{V}$, we have for every $\varepsilon \in (0, \frac{\sigma}{L}]$, letting*

$$B = \Theta\big(\tfrac{1}{\varepsilon^2}\big) \quad , \quad T = \Theta\Big(\tfrac{L^{2/3}\sigma^{1/3}}{\varepsilon^{10/3}}\Big) \quad and \quad \alpha = \Theta\Big(\tfrac{\varepsilon^{4/3}}{\sigma^{1/3}L^{2/3}}\Big)$$

*we have that* `Natasha1.5`*$(f, x^{\varnothing}, B, T/B, \alpha)$ outputs a point $x^{\mathsf{out}}$ with $\mathbb{E}[\|\nabla f(x^{\mathsf{out}})\|] \leq \varepsilon$, and needs $O(T)$ computations of stochastic gradients. (See also Figure 3(b).)*

We emphasize that the additional factor $\sigma^{1/3}$ in the numerator of $T$ shall become our key to achieve faster algorithm for finding approximate local minima in Section 4. Also, if the requirement $\varepsilon \leq \frac{\sigma}{L}$ is not satisfied, one can replace $\sigma$ with $\varepsilon L$; accordingly, $T$ becomes $O\big(\frac{L}{\varepsilon^3} + \frac{L^{2/3}\sigma^{1/3}}{\varepsilon^{10/3}}\big)$

We note that the SGD4 method of [5] (which appeared after this paper) achieves $T = O\big(\frac{L}{\varepsilon^2} + \frac{\sigma}{\varepsilon^4}\big)$. It is better than `Natasha1.5` only when $\sigma \leq \varepsilon L$. We compare them in Figure 3(b), and emphasize that it is necessary to use `Natasha1.5` (rather than SGD4) to design `Natasha2` of the next section.

**Extension.** In fact, we show Theorem 1 in a more general *proximal* setting. That is, to minimize $F(x) := f(x) + \psi(x)$ where $\psi(x)$ is proper convex function that can be *non-smooth*. For instance, if $\psi(x)$ is the indicator function of a convex set, then Problem (1.1) becomes constraint minimization; and if $\psi(x) = \|x\|_1$, we encourage sparsity. At a first reading of its proof, one can assume $\psi(x) \equiv 0$.

## 3.2 Our Intuition

We first recall the main idea of the SVRG method [28, 48], which is an *offline* algorithm. SVRG divides iterations into epochs, each of length $n$. It maintains a snapshot point $\widetilde{x}$ for each epoch, and computes the full gradient $\nabla f(\widetilde{x})$ only for snapshots. Then, in each iteration $t$ at point $x_t$, SVRG defines gradient estimator $\widetilde{\nabla} f(x_t) := \nabla f_i(x_t) - \nabla f_i(\widetilde{x}) + \nabla f(\widetilde{x})$ which satisfies $\mathbb{E}_i[\widetilde{\nabla} f(x_t)] = \nabla f(x_t)$, and performs proximal update $x_{t+1} \leftarrow x_t - \alpha\widetilde{\nabla} f(x_t)$ for learning rate $\alpha$.

For minimizing non-convex functions, SVRG does not take advantage of parameter $\sigma$ *even if* the learning rate can be adapted to $\sigma$. This is because SVRG (and in fact SGD and GD too) rely on gradient-descent analysis to argue for objective decrease *per iteration*. This is blind to $\sigma$.[9]

The prior work `Natasha1` takes advantage of $\sigma$. `Natasha1` is similar to `SVRG`, but it further divides each epoch into sub-epochs, each with a starting vector $\widehat{x}$. Then, it replaces $\widetilde{\nabla} f(x_t)$ with $\widetilde{\nabla} f(x_t) + 2\sigma(x_t - \widehat{x})$. This is equivalent to replacing $f(x)$ with $f(x) + \sigma \|x - \widehat{x}\|^2$, where the center $\widehat{x}$ changes every sub-epoch. We view this additional term $2\sigma(x_t - \widehat{x})$ as a type of _retraction_. Conceptually, it stabilizes the algorithm by moving a bit in the backward direction. Technically, it enables us to perform only mirror-descent type of analysis, and thus bypass the issue of `SVRG`.

**Our Algorithm.** Both `SVRG` and `Natasha1` are offline methods, because the gradient estimator requires the full gradient computation $\nabla f(\widetilde{x})$ at snapshots $\widetilde{x}$. A natural fix —originally studied by practitioners but first formally analyzed by Lei et al. [30]— is to replace the computation of $\nabla f(\widetilde{x})$ with $\frac{1}{|S|} \sum_{i \in S} \nabla f_i(\widetilde{x})$, for a random batch $S \subseteq [n]$ with fixed cardinality $B := |S| \ll n$. This allows us to shorten the epoch length from $n$ to $B$, thus turning `SVRG` and `Natasha1` into _online_ methods.

How large should we pick $B$? By Chernoff bound, we wish $B \approx \frac{1}{\varepsilon^2}$ because our desired accuracy is $\varepsilon$. One can thus _hope_ to replace the parameter $n$ in the complexities of `SVRG` and `Natasha1.5` (ignoring the dependency on $L$):

$$T = O\big(n + \frac{n^{2/3}}{\varepsilon^2}\big) \quad \text{and} \quad T = O\big(n + \frac{n^{1/2}}{\varepsilon^2} + \frac{\sigma^{1/3} n^{2/3}}{\varepsilon^2}\big)$$

with $B \approx \frac{1}{\varepsilon^2}$. This "wishful thinking" gives

$$T = O\big(\varepsilon^{-10/3}\big) \quad \text{and} \quad T = O\big(\varepsilon^{-3} + \sigma^{1/3} \varepsilon^{-10/3}\big).$$

These are exactly the results achieved by `SCSG` [30] and to be achieved by our new `Natasha1.5`.

_Unfortunately,_ Chernoff bound itself is not sufficient in getting such rates. Let

$$\mathbf{e} := \frac{1}{|S|} \sum_{i \in S} \nabla f_i(\widetilde{x}) - \nabla f(\widetilde{x})$$

denote the bias of this new gradient estimator, then when performing iterative updates, this bias $\mathbf{e}$ gives rise to two types of error terms: "first-order error" terms —of the form $\langle \mathbf{e}, x - y \rangle$— and "second-order error" term $\|\mathbf{e}\|^2$. Chernoff bound ensures that the second-order error $\mathbb{E}_S[\|\mathbf{e}\|^2] \leq \varepsilon^2$ is bounded. However, first-order error terms are the true bottlenecks.

In the offline method `SCSG`, Lei et al. [30] carefully performed updates so that all "first-order errors" cancel out. To the best of our knowledge, this analysis cannot take advantage of $\sigma$ even if the algorithm knows $\sigma$. (Again, for experts, this is because `SCSG` is based on gradient-descent type of analysis but not mirror-descent.)

In `Natasha1.5`, we use the aforementioned retraction to ensure that all points in a single sub-epoch are close to each other (based on mirror-descent type of analysis). Then, we use Young's inequality to bound $\langle \mathbf{e}, x - y \rangle$ by $\frac{1}{2}\|\mathbf{e}\|^2 + \frac{1}{2}\|x - y\|^2$. In this equation, $\|\mathbf{e}\|^2$ is already bounded by Chernoff concentration, and $\|x - y\|^2$ can also be bounded as long as $x$ and $y$ are within the same sub-epoch. This summarizes the high-level technical contribution of `Natasha1.5`.

We formally state `Natasha1.5` in Algorithm 1, and it uses big-$O$ notions to hide dependency in $L$, $\Delta_f$, and $\mathcal{V}$. The more general code to take care of the proximal term is in Algorithm 3 of Section 6.

# 4 Natasha 2: Finding Approximate Local Minima

Stochastic gradient descent (SGD) find approximate local minima [22], under (A1), (A2) and an additional assumption (A4):

$f(x)$ is second-order $L_2$-Lipschitz smooth, meaning $\|\nabla^2 f(x) - \nabla^2 f(y)\|_2 \leq L_2 \cdot \|x - y\|$.
(A4)

_Remark_ 4.1. (A4) is necessary to make the task of find appx. local minima meaningful, for the same reason Lipschitz smoothness was needed for finding stationary points.

**Definition 4.2.** _We say $x$ is an $(\varepsilon, \delta)$-approximate local minimum of $f(x)$ if_[10]

$$\|\nabla f(x)\| \leq \varepsilon \quad and \quad \nabla^2 f(x) \succeq -\delta \mathbf{I} \;,$$

---

bounded nonconvexity parameter of $f(x)$. For readers interested in the difference between gradient and mirror descent, see [11].

[10]The notion "$\nabla^2 f(x) \succeq -\delta \mathbf{I}$" means all the eigenvalues of $\nabla^2 f(x)$ are above $-\delta$.

*or $\varepsilon$-approximate local minimum if it is $(\varepsilon, \varepsilon^{1/C})$-approximate local minimum for constant $C \geq 1$.*

Before our work, Ge et al. [22] is the only result that gives provable *online* complexity for finding approximate local minima. Other previous results, including `SVRG`, `SCSG`, `Natasha1`, and even `Natasha1.5`, do not find approximate local minima and may be stuck at saddle points.[11] Ge et al. [22] showed that, hiding factors that depend on $L$, $L_2$ and $\mathcal{V}$, SGD finds an $\varepsilon$-approximate local minimum of $f(x)$ in gradient complexity $T = O(\mathsf{poly}(d)\varepsilon^{-4})$. This $\varepsilon^{-4}$ factor seems necessary since SGD needs $T \geq \Omega(\varepsilon^{-4})$ for just finding stationary points (see Appendix B and Table 1).

*Remark* 4.3. Offline methods are often studied under $(\varepsilon, \varepsilon^{1/2})$-approximate local minima. In the online setting, Ge et al. [22] used $(\varepsilon, \varepsilon^{1/4})$-approximate local minima, thus giving $T = O\left(\frac{\mathsf{poly}(d)}{\varepsilon^4} + \frac{\mathsf{poly}(d)}{\delta^{16}}\right)$. In general, it is better to treat $\varepsilon$ and $\delta$ separately to be more general, but nevertheless, $(\varepsilon, \varepsilon^{1/C})$-approximate local minima are always better than $\varepsilon$-approximate stationary points.

## 4.1 Our Theorem

We propose a new method `Natasha2`<sup>full</sup> which, very informally speaking, alternatively

- finds approximate stationary points of $f(x)$ using `Natasha1.5`, or
- finds negative curvature of the Hessian $\nabla^2 f(x)$, using Oja's online eigenvector algorithm.

In this section, we define gradient complexity $T$ to be the number of stochastic gradient computations plus Hessian-vector products. Let $\Delta_f$ be any upper bound on $f(x_0) - f(x^*)$ where $x_0$ is the starting point. In this section, to present the simplest results, we use the big-$O$ notion to hide dependency in $L$, $L_2$, $\Delta_f$, and $\mathcal{V}$. In Section 7, we shall add back such dependency for a more general description of the algorithm. Our main result can be stated as follows:

**Theorem 2** (informal). *Under (A1), (A2) and (A4), for any $\varepsilon \in (0, 1)$ and $\delta \in (0, \varepsilon^{1/4})$, `Natasha2`$(f, y_0, \varepsilon, \delta)$ outputs a point $x^{\mathsf{out}}$ so that, with probability at least 2/3:*

$$\|\nabla f(x^{\mathsf{out}})\| \leq \varepsilon \quad and \quad \nabla^2 f(x^{\mathsf{out}}) \succeq -\delta \mathbf{I} \ .$$

*Furthermore, its gradient complexity is $T = \widetilde{O}\left(\frac{1}{\delta^5} + \frac{1}{\delta \varepsilon^3}\right)$.*[12]

*Remark* 4.4. If $\delta > \varepsilon^{1/4}$ we can replace it with $\delta = \varepsilon^{1/4}$. Therefore, $T = \widetilde{O}\left(\frac{1}{\delta^5} + \frac{1}{\delta \varepsilon^3} + \frac{1}{\varepsilon^{3.25}}\right)$.

*Remark* 4.5. The follow-up work [10] replaced Hessian-vector products in `Natasha2` with only stochastic gradient computations, turning `Natasha2` into a pure first-order method. That paper is built on ours and thus all the proofs of this paper are still needed.

**Corollary 4.6.** *$T = \widetilde{O}(\varepsilon^{-3.25})$ for finding $(\varepsilon, \varepsilon^{1/4})$-approximate local minima. This is better than $T = O(\varepsilon^{-10/3})$ of SCSG for finding only $\varepsilon$-approximate stationary points.*

**Corollary 4.7.** *$T = \widetilde{O}(\varepsilon^{-3.5})$ for finding $(\varepsilon, \varepsilon^{1/2})$-approximate local minima. This was not known before and is matched by several follow-up works using different algorithms [5, 10, 44, 46].*

## 4.2 Our Intuition

It is known that the problem of finding $(\varepsilon, \delta)$-approximate local minima, at a high level, "reduces" to (repeatedly) finding $\varepsilon$-approximate stationary points for an $O(\delta)$-nonconvex function [1, 13]. Specifically, Carmon et al. [13] proposed the following procedure. In every iteration at point $y_k$, detect whether the minimum eigenvalue of $\nabla^2 f(y_k)$ is below $-\delta$:

- if yes, find the minimum eigenvector of $\nabla^2 f(y_k)$ approximately and move in this direction.
- if no, let $F^k(x) := f(x) + L\left(\max\left\{0, \|x - y_k\| - \frac{\delta}{L_2}\right\}\right)^2$, which can be proven as $5L$-smooth and $3\delta$-nonconvex; then find an $\varepsilon$-approximate stationary point of $F^k(x)$ to move there. Intuitively, $F^k(x)$ penalizes us from moving out of the "safe zone" of $\left\{x \colon \|x - y_k\| \leq \frac{\delta}{L_2}\right\}$.

**Algorithm 2** `Natasha2`$(f, y_0, \varepsilon, \delta)$

**Input:** function $f(x) = \frac{1}{n}\sum_{i=1}^{n} f_i(x)$, starting vector $y_0$, target accuracy $\varepsilon > 0$ and $\delta > 0$.

1: **if** $\frac{\varepsilon^{1/3}}{\delta} \geq 1$ **then** $\widetilde{L} = \widetilde{\sigma} \leftarrow \Theta(\frac{\varepsilon^{1/3}}{\delta}) \geq 1;$        ⋄ *the boundary case for large $L_2$*

2: **else** $\widetilde{L} \leftarrow 1$ and $\widetilde{\sigma} \leftarrow \Theta(\frac{\varepsilon}{\delta^3}) \in [\delta, 1].$

3:   $X \leftarrow [];$

4: **for** $k \leftarrow 0$ **to** $\infty$ **do**

5:     Apply Oja's algorithm to find minEV $v$ of $\nabla^2 f(y_k)$ for $\widetilde{\Theta}(\frac{1}{\delta^2})$ iterations

                                                    ⋄ *see Lemma 5.3*

6:     **if** $v \in \mathbb{R}^d$ is found s.t. $v^\top \nabla^2 f(y_k) v \leq -\frac{\delta}{2}$ **then**

7:         $y_{k+1} \leftarrow y_k \pm \frac{\delta}{L_2} v$ where the sign is random.

8:     **else**                                 ⋄ *it satisfies $\nabla^2 f(y_k) \succeq -\delta \mathbf{I}$*

9:         $F^k(x) := f(x) + L(\max\{0, \|x - y_k\| - \frac{\delta}{L_2}\})^2.$

10:        run `Natasha1.5`$\big(F^k, y_k, \Theta(\varepsilon^{-2}), 1, \Theta(\varepsilon\delta)\big).$    ⋄ *$F^k(\cdot)$ is $\widetilde{L}$-smooth and $\widetilde{\sigma}$-nonconvex*

11:        let $\widehat{y}_k, y_{k+1}$ be the vector $\widehat{y}$ and $\widehat{x}$ when Line 13 is reached in `Natasha1.5`.

12:        $X \leftarrow [X, (y_k, \widehat{y}_k)];$

13:        break the for loop if have performed $\Theta(\frac{1}{\delta\varepsilon})$ first-order steps.

14:     **end if**

15: **end for**

16: $(y, \widehat{y}) \leftarrow$ a random pair in $X$.                 ⋄ *in practice, simply output $\widehat{y}_k$*

17: define convex function $g(x) := f(x) + L(\max\{0, \|x - y\| - \frac{\delta}{L_2}\})^2 + \widetilde{\sigma}\|x - \widehat{y}\|^2.$

18: use SGD to minimize $g(x)$ for $\widetilde{\Theta}(\frac{1}{\varepsilon^2})$ steps and output $x^{\mathsf{out}}$.

---

Previously, it was thought necessary to achieve high accuracy for both tasks above. This is why researchers have only been able to design offline methods: in particular, the shift-and-invert method [21] was applied to find the minimum eigenvector, and `repeatSVRG` was applied to find a stationary point of $F^k(x)$.[13]

In this paper, we apply efficient *online* algorithms for the two tasks: namely, Oja's algorithm (see Section 5.1) for finding minimum eigenvectors, and our new `Natasha1.5` algorithm (see Section 3.2) for finding stationary points. More specifically, for Oja's, we only decide if there is an eigenvalue below threshold $-\delta/2$, or conclude that the Hessian has all eigenvalues above $-\delta$. This can be done in an online fashion using $O(\delta^{-2})$ Hessian-vector products (with high probability) using Oja's algorithm. For `Natasha1.5`, we only apply it for a single epoch of length $B = \Theta(\varepsilon^{-2})$. Conceptually, this shall make the above procedure online and run in a complexity independent of $n$.

*Unfortunately,* technical issues arise in this "wishful thinking."

Most notably, the above process finishes only if `Natasha1.5` finds an approximate stationary point $x$ of $F^k(x)$ that is *also* inside the safe zone $\left\{x \colon \|x - y_k\| \leq \frac{\delta}{L_2}\right\}$. This is because $F^k(x) = f(x)$ inside the safe zone and therefore $\|\nabla F^k(x)\| \leq \varepsilon$ also implies $\|\nabla f(x)\| \leq 2\varepsilon$.

What can we do if we move out of the safe zone? To tackle this case, we show an additional property of `Natasha1.5` (see Lemma 6.5). That is, the amount of objective decrease —i.e., $f(y_k) - f(x)$ if $x$ moves out of the safe zone— must be proportional to the distance $\|x - y_k\|^2$ we travel in space. Therefore, if $x$ moves out of the safe zone, then we can decrease sufficiently the objective. This is also a good case. This summarizes some high-level technical ingredient of `Natasha2`.

We formally state `Natasha2` in Algorithm 2, and it uses the big-$O$ notion to hide dependency in $L$, $L_2$, $\mathcal{V}$ and $\Delta_f$. The more general code to take care of all the parameters can be found in Algorithm 5 of Section 7.

Finally, we stress that although we borrowed the construction of $f(x) + L\big(\max\big\{0, \|x - y_k\| - \frac{\delta}{L_2}\big\}\big)^2$ from the offline algorithm of Carmon et al. [13], our `Natasha2` algorithm and analysis are different from them in all other aspects.

## Footnotes

[4]In practice, stochastic gradients naturally incur "random noise" and adding perturbation may not be needed.

[5]As we shall see, smoothness is necessary for finding approximate local minima with provable guarantees.

[6] Previous authors also refer to this notion as "approximate convex", "almost convex", "hypo-convex", "semi-convex", or "weakly-convex." We call it $\sigma$-nonconvex to stress the point that $\sigma$ can be as large as $L$ (any $L$-smooth function is automatically $L$-nonconvex).

[7] Historically, in first-order literatures, $x$ is called $\varepsilon$-approximate if $\|\nabla f(x)\|^2 \leq \varepsilon$; in second-order literatures, $x$ is $\varepsilon$-approximate if $\|\nabla f(x)\| \leq \varepsilon$. We adapt the latter notion following Polyak and Nesterov [34, 36].

[8]For instance, if the variance $\mathcal{V}$ is unbounded, we cannot even tell if a point $x$ satisfies $\|\nabla f(x)\| \leq \varepsilon$ using finite samples. Also, if $f(x)$ is not Lipschitz smooth, it may contain sharp turning points (e.g., behaves like absolute value function $|x|$); in this case, finding $\|\nabla f(x)\| \leq \varepsilon$ can be as hard as finding $\|\nabla f(x)\| = 0$, and is NP-hard in general.

[9] These results argue for objective decrease per iteration, of the form $f(x_t) - f(x_{t+1}) \geq \frac{\alpha}{2}\|\nabla f(x_t)\|^2 - \frac{\alpha^2 L}{2}\mathbb{E}\big[\|\nabla f(x_t) - \widetilde{\nabla} f(x_t)\|^2\big]$. Unlike mirror-descent analysis, this inequality cannot take advantage of the

[11]These methods are based on the "variance reduction" technique to reduce the random noise of SGD. They have been criticized by practitioners for performing poorer than SGD on training neural networks, because the noise of SGD allows it to escape from saddle points. Variance-reduction based methods have less noise and thus cannot escape from saddle points.

[12]Throughout this paper, we use the $\widetilde{O}$ notion to hide at most one logarithmic factor in all the parameters (namely, $n, d, L, L_2, \mathcal{V}, 1/\varepsilon, 1/\delta$).

[13] `repeatSVRG` is an offline algorithm, and finds an $\varepsilon$-approximate stationary point for a function $f(x)$ that is $\sigma$-nonconvex. It is divided into stages. In each stage $t$, it considers a modified function $f_t(x) := f(x) + \sigma\|x - x_t\|^2$, and then apply the accelerated `SVRG` method to minimize $f_t(x)$. Then, it moves to $x_{t+1}$ which is a sufficiently accurate minimizer of $f_t(x)$.

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
