[Reviews · NeurIPS 2018]

Reviewer 1



This submission considers an online optimization problem, where the underlying objective function is either an expectation or a finite sum of functions. It is assumed that the objective is nonconvex, and the goal of the submission is to develop an efficient online algorithm (that is, a method with a complexity that is independent on the number of components defining the objective) that finds an approximate local minimum, i. e. a point at which the gradient norm is less than $\epsilon$ and the minimum Hessian eigenvalue is at least $-\delta$, where $\epsilon$ and $\delta$ are two positive thresholds. The development of online algorithms is extremely relevant in the context of machine learning, in that it reduces the dependence of the method to the size of the available data. Considering nonconvex problems is also of critical importance, particularly given the outbreak of deep learning formulations. The submission appears to be written in order to help the reader follow the reasoning that lead to the derivation of the new method and results. The introductory section conveys in my opinion the right ideas, for its most part. I am concerned by the comparison made on lines 71-76: it seems that the best guarantees obtained in references [35], [7,39] and this submission correspond to different choices for the second-order tolerance $\delta$, namely, $\delta=\epsilon^{1/2}$ for [35] (and [39] for a bound in $\epsilon^{-3.5}$), $\delta=\epsilon^{4/9}$ for [39] and a bound in $\epsilon^{-3.333}$, and $\delta=\epsilon^{1/4}$ for [7] and this submission, with bounds respectively in $\epsilon^{-3.333}$ and $\epsilon^{-3.25}$ Adding this precision would help in comparing those methods, especially given that some of the cited references study multiple algorithms (this is also valid for Table 1 in appendix, to a lesser extent). The core of the submission begins in Section 3 with the description of the Natasha 1.5 method to find approximate stationary points, at which the gradient has small norm. This method is an extension of the original Natasha algorithm, adapted to the online setting by exploiting a so-called strong nonconvexity" property of the function. Certain aspects and remarks of this section can be difficult to understand without resorting to the appendix: in particular, the use of a proximal operator (and Definition 2.2) may be deferred to the appendix, and more simply mentioned, in order to focus on the exploitation of the strong nonconvexity property, and its main difference with the original Stochastically Controlled Stochastic Gradient (SCSG) paradigm. The main algorithm, Natasha 2, is detailed in Section 4. The conditions $\delta \in (0, \epsilon^{1/4})$ in Theorem 2 and $\delta \ge \epsilon^{1/4}$ in Section 1 (or $\delta > \epsilon^{1/4}$ in Remark 4.4) seem mutually exclusive, and I believe the presentation would have been simplified by considering only the special case $\delta = \epsilon^{1/4}$. Referring to the appendix is also helpful here to understand what is behind these informal theorems, particular toward the end of the section as several technical difficulties are sketched. Overall, even if the results and the analysis appear correct to me, it is my opinion that the presentation of the submission can prevent a reader from fully appreciating the quality of its results. More specifically, the novelty of the Natasha 2 algorithm, that allows for the derivation of a bound in $\epsilon^{-3.25}$, appears hidden among the various considerations mentioned in the submission, including the fact that this bound is not necessarily better than those in $\epsilon^{-3.5}$. Moreover, the submission relies heavily on its supplementary material, which tends to indicate that the main body is not sufficient to get a good idea of this work. I strongly believe that the results of the submission are sufficiently original and significant: it is clear that they have inspired follow-up works, and in that sense they fully deserve publication. However, it does not seem that the NIPS format is the most appropriate for exposing the method, and in my opinion, it would not do justice to the efforts made by the authors to position their work within the existing literature. For these reasons, I regretfully give this submission an overall score of 4 for my initial review. Additional comments: a) Some sentences are oddly formulated, for instance: - Being easy to implement, however, we raise two main efficiency issues" line 33; - any point close to a saddle point must have a negative curvature" lines 53-54. b) The derivation of the series of equations lines 674-675 could be more detailed. There is also an extra $p$ on the last three equations of line 674, as the term is in the sum. Edit of the review following author feedback: I am pleased that the author took my review into consideration, and I concur with his responses. In my opinion, addressing these concerns was necessary to properly position this work into the literature (and acknowledge its impact on later works). I trust the author to make the necessary changes to his submission so as to address these concerns, and correct the typos that I pointed out. One exception is my comment about simplifying the presentation by choosing $\delta=\epsilon^{1/4}$ throughout the submission. I understand the author's point in the feedback, and although I do believe it would make a camera-ready version easier to understand, I will not insist on it. The author should feel free to maintain the use of $\delta$. Even though it seems to me that this submission has grown out beyond a NIPS paper ever since its first appearance online, I am willing to change my assessment and recommend acceptance of the submission, following the other reviewers' assessments.

Reviewer 2



This paper proposes a novel method to find local minima of smooth non-convex functions. Specifically, the proposed method utilizes the Hessian information to help avoid saddle points. Overall, the idea is novel and it is presented very clearly. My only question is how to set the parameter \sigma in Algorithm 1 and L_2 in Algorithm 2 in practical applications. For feedback: In the feedback, the author has given a method to select these two parameters. So, I will keep my rating. At last, it is recommended that the author can discuss how to select these parameters in the final version.

Reviewer 3



The paper discusses a faster algorithm for finding local minima in non-convex problems. The authors provide theoretical guarantees. They argue that as a byproduct thier rigorous results could help in desiging novel algorithmic schemes. This is a solid paper, incremental but still worth accepting (in my opinion). Other manuscripts provide almost equivalent results.